# Are Dietary Habits the Missing Link Between Hashimoto’s Thyroiditis and Osteoporosis?

**DOI:** 10.3390/nu17132109

**Published:** 2025-06-25

**Authors:** Anita Vergatti, Veronica Abate, Francesca Garofano, Antonella Fiore, Gianpaolo De Filippo, Pasquale Strazzullo, Domenico Rendina

**Affiliations:** 1Department of Clincal Medicine and Surgery, University of Naples Federico II, 80131 Naples, Italy; anita.vergatti@unina.it (A.V.); veronica.abate@unina.it (V.A.); garofanofran.1@gmail.com (F.G.); fioreantonella95@gmail.com (A.F.); 2Assistance Publique-Hôpitaux de Paris, Hôpital Robert Debré, Service d’Endocrinologie et Diabétologie, 75019 Paris, France; gianpaolo.defilippo@aphp.fr; 3Internal Medicine, University of Naples Federico II, 80131 Naples, Italy; strazzul@unina.it

**Keywords:** osteoporosis, Hashimoto’s thyroiditis, dietary habits, bone health, Mediterranean diet

## Abstract

Bone metabolism is a dynamic process involving continuous bone formation and resorption, orchestrated by the interplay between osteoblasts and osteoclasts. Osteoporosis (Op), the most prevalent osteo-metabolic disorder globally, results from an imbalance in this remodeling cycle. Hashimoto’s thyroiditis (HT), a chronic autoimmune thyroid disorder, has been increasingly recognized as a contributor to bone loss, even in euthyroid individuals. HT is marked by immune dysregulation, autoantibody production, and chronic inflammation, factors that can alter bone remodeling. Furthermore, both thyroid-stimulating hormone (TSH) and thyroid hormones (THs) independently influence bone health. Low TSH and elevated TH levels, including in subclinical states, have been linked to reduced bone mineral density (BMD) and increased fracture risk. Nutritional factors, particularly selenium and iodine intake, modulate both thyroid and bone function, and can be considered as a link between HT and Op. In particular, antioxidant-rich diets such as the Mediterranean diet may confer protective effects. This review integrates current clinical and experimental evidence linking HT with bone metabolism disorders, emphasizing the multifactorial nature of bone fragility in autoimmune thyroid disease and the potential role of diet in mitigating its impact.

## 1. Introduction: Bone and Hashimoto’s Thyroiditis

Osteoporosis (Op) is the most common osteo-metabolic disorder worldwide, affecting over 200 million people [1]. The incident of Op increases with age, and its prevalence is higher in women than men [1,2]. Op is associated with an imbalance between bone resorption and formation, leading to a progressive deterioration of bone microarchitecture and bone mechanical properties, ultimately increasing the risk of fragility fractures [3]. Recent clinical and experimental evidence highlights the complex interaction between the bone and immune systems [4]. The immune and skeletal systems influence each other during cell activation, proliferation, and cellular senescence [4]. Macrophages enhance osteoblastogenesis via interleukin 18 (IL18) [5], while T cells influence osteoclastogenesis through IL1, IL6, IL4, and interferon-γ (IFN-γ) [6,7]. On the other hand, the receptor activator of nuclear factor-κB (RANK)–RANK Ligand (RANKL)–osteoprotegerin (OPG) axis, a signaling pathway that plays a crucial role in bone metabolism, is also involved in the regulation of immune cells. RANKL, a member of the tumor necrosis factor-α (TNF-α) superfamily, typically binds to RANK expressed on osteoclasts contributing to bone resorption. RANKL–RANK signaling also promotes the activation and survival of dendritic cells and T-cells, enhancing the immune response, and takes a part in B-cell recruitment and follicle organization [8].

Based on this knowledge, Op and other osteo-metabolic disorders should be considered clinical conditions also affected by immune system disfunction [9]. Chronic inflammation accelerates bone loss and increases fracture risks, underscoring the detrimental effects of prolonged immune activation on skeletal integrity [10].

Hashimoto’s thyroiditis (HT) represents a prototypical autoimmune disorder, with an immunological phenotype that could affect skeletal development and bone homeostasis. HT represents the most common cause of hypothyroidism in developed countries with adequate dietary iodine intake (i.e., median urinary iodine excretion ≥ 100 μg/L). It affects about 160 million people worldwide [11,12,13], predominantly adult women, with a female-to-male ratio of 10:1 [11]. HT is characterized by lymphocytic infiltration, varying degrees of thyroid disfunction, circulating antibodies against thyroid antigens, i.e., autoantibodies against thyroglobulin (TgAb) and thyroid peroxidase (TPOAb), and thyroid enlargement (i.e., goiter).

HT is a recognized multisystemic disorder, which affects several organ systems, including the neurological, cardiovascular, dermatological, gastrointestinal, and musculoskeletal systems. A deeper understanding of the systemic nature of Hashimoto’s disease may unlock innovative therapeutic strategies that move beyond standard hormone replacement, aiming instead to modulate the fundamental autoimmune mechanisms driving the disorder [14].

Based on the available literature, it is evident that both Op and HT share common pathogenic mechanisms, such as oxidative stress and unhealthy dietary habits. On the one hand, oxidative stress, resulting from an imbalance between pro-oxidant and antioxidant agents, affects both bone and thyroid health. On the other hand, the influence of a healthy lifestyle, and, in particular, an appropriate dietary patterns, may offer new insights into the pathophysiology of both disorders and contribute to the development of novel therapeutic strategies.

Given the critical role of the thyroid in bone homeostasis [15,16] and the significant socio-economic impact of both conditions, the aim of this review is to explore and clarify the influence of HT and its related components on bone health, and to evaluate the potential role of dietary habits as a possible, yet often overlooked, link between these two pathological states.

## 2. The Role of Thyroid-Stimulating Hormone and Thyroid Hormones in Bone Health

Thyroid-stimulating hormone (TSH), also known as thyrotropin, is a pituitary glycoprotein hormone that stimulates the release of THs, primarily thyroxine (T4), which is then converted into triiodothyronine (T3) [17]. THS role in bone health remains debated. Different studies have analyzed the effect of subclinical hyperthyroidism and hypothyroidism on bone mineral density (BMD). Mazziotti et al. (2010) [18] demonstrated that low–normal TSH levels were associated with a high prevalence of vertebral fractures (35%) in postmenopausal women, independently of THs, age, and BMD. Also, high–normal TSH levels appear to be associated with a reduced risk of non-vertebral fractures (reduced by 35%) in postmenopausal women over a 6-year follow-up [18,19]. Leader et al. (2014) confirmed that low TSH levels, even within the normal range (0.5–5.5 mIU/L), were associated with an increased risk of hip fractures in women [Odds Ratio (OR) 1.28, 95% CI (1.03–1.59)] but not in men [20]. In contrast, another study found no significant association between TSH levels within the normal range and changes in BMD, suggesting that TSH may not directly influence bone mass in euthyroid individuals, also after adjustment for height, weight, age, smoking, physical activity, and, for postmenopausal females, the use of hormonal replacement therapy [21]. These conflicting findings point to potential size and sex-specific effects.

To summarize the role of TSH on bone health, Segna et al. demonstrated that individuals with subclinical hyperthyroidism experienced greater femoral neck bone loss (−0.18% ΔBMD) compared to those with euthyroid function, whereas subclinical hypothyroidism was not associated with bone loss [22]. Bauer et al. (2001) reported an increased risk of hip [relative hazard (RH) 3.6] and vertebral fractures (OR 4.5, 95% CI 1.3, 15.6) in women over 65 years of age with low serum TSH levels, independently of hyperthyroidism [23]. Abrahamsen et al. (2014) found a higher risk of hip fractures and all main osteoporotic fractures in individuals with low TSH and normal T4 and T3 [24]. Finally, a meta-analysis including 70,298 subjects showed that subclinical hyperthyroidism was associated with an increased risk of hip [Hazard Ratio (HR) 1.61] and other fractures (HR 1.98), particularly among those with TSH levels below 0.10 mIU/L with subclinical hypothyroidism [25].

Demonstrating the independent role of TSH alterations in bone health is a challenge. Abe et al. (2003) [26] provided evidence for direct effects of TSH on bone formation and bone resorption. These effects were mediated by the TSH receptor (TSHR). Indeed, haplo-insufficient TSHR mice showed bone loss and osteopenia [26]. Baliram et al. demonstrated that the absence of TSH signaling increased bone loss. Indeed, TSHR-Knockout mice with hyperthyroidism had a higher bone resorption rate compared to wild-type mice [27]. TSH seems to inhibit osteoclast formation and increase osteoblast differentiation [28,29], even if the exact mechanisms are not fully understood.

THs play a key role in skeletal growth and maturation [30]. Children with undiagnosed severe hypothyroidism show delayed skeletal development, defective endochondral ossification, and short stature [31]. In adulthood, excess THs cause high bone turnover-mediated bone loss, while deficiency causes low bone turnover-mediated bone loss due to reduced osteoclastic bone resorption and decreased osteoblastic activity [31,32]. T3 stimulates direct and indirect osteoblast proliferation and differentiation through various growth factors and cytokines, and by activating MAPK signaling and the Wnt pathway [33]. Moreover, T3 appears to have a role in osteoblast response to parathormone (PTH) by modulating the expression of the PTH- and PTH related peptide-receptor [34]. On the contrary, it is not clear whether T3 acts directly on osteoclasts or indirectly by stimulating osteoblasts, osteocytes, or other bone cells [35,36]. Van der Deure et al. (2008) demonstrated that free T4 (FT4) was negatively associated with bone parameters and BMD: lumbar spine BMD β = −0.003, femoral neck BMD β = −0.005, and cortical thickness β = −0.001 [37]. Murphy et al. (2010) [19] confirmed these results, showing that higher FT4 (β = −0.091) and FT3 (β = −0.087) were associated with lower BMD at the hip, and higher FT4 was associated with increased bone loss at the hip (β = −0.09), also after adjustment for age, body mass index, and BMD. Moreover, the risk of non-vertebral fracture was increased by 20% and 33% in women with higher FT4 and FT3, respectively. On the contrary, it was reduced by 35% in case of higher TSH [18]. To clarify the causal relationship between thyroid disorders and Op, Liu et al. (2025) used a two-sample Mendelian randomization analysis proving that TSH mediated 5.31% of the association between hypothyroidism and Op, while FT4 mediated 9.67% of the relationship between hyperthyroidism and Op [38]. In contrast, Leng et al. (2024) [39] investigated the causal relationships between hypothyroidism, hyperthyroidism, FT3, FT4, TSH, and OP risk, and found no significant causal associations between FT3, FT4, TSH, and the risk of developing osteoporosis. On the other hand, hypothyroidism may elevate the risk of osteoporosis (OR 1.082) by altering blood metabolite levels, such as triglycerides [39].

Appendix A summarizes studies on the role of TSH in bone health [18,19,20,21,22,23,24,25,37,38,39].

## 3. Determinants of Autoimmune Thyroid Disease and Their Impact on Bone Health

HT directly impacts bone health, independently of hormonal dysfunction [40,41]. TgAb, TPOAb, and some cytokines dysregulated in HT individuals, such as IL1β, IL6, IL17, IL23, tumor necrosis factor (TNF)-α, and IFN–γ, are key pathogenic factors contributing to BMD alterations [42,43].

Lambrinoudaki et al. (2017) evaluated the association between TPOAb and TgAb with BMD in euthyroid postmenopausal women, demonstrating a higher risk of fractures in individuals with TPOAb and TgAb (OR 3.308 and 6.328, respectively) [44]. Polovina et al. (2017) [45] showed that TPOAb is a potential marker of higher fracture risk in euthyroid postmenopausal women. TPOAb were found to be strongly linked with an increased risk of any type of fracture, independently of TSH values (OR 7.8) [45]. Indeed, elevated TPOAb levels have been associated with higher Fracture Risk Assessment Tool (FRAX) score, indicating an increased risk of fractures. In particular, the FRAX score was significantly higher in the group with subclinical hypothyroidism than in the controls (6.5 vs. 4.35) [46].

IL-6 seems to accelerate bone turnover by inhibiting bone formation and promoting bone reabsorption [47]. In particular, a meta-analysis conducted by Chen et al. showed a higher Op risk in IL-6 572C/G additive (OR 2.25), dominant (OR 1.42), and recessive (OR 1.96) models [48]. Moreover, IL-6 produces increased RANKL mRNA expression [49] and influences bone mass through glycoprotein 130-STAT 1/3 signaling pathway [50].

Additionally, high IL17 levels were associated with low BMD in postmenopausal women [51], but its role in osteoblastogenesis remains to be further defined. Zhang et al. (2017) reported that IL17 inhibited osteoblast differentiation and bone regeneration in rats [52]. In contrary, Kim et al. (2020) showed that IL17 stimulated osteoblast differentiation and osteoblast-dependent osteoclastogenesis in vitro [53].

The IL-23/IL-17 axis plays a significant role in the pathogenesis of HT [54] as well as in systemic bone loss [55]. IL23 can indirectly inhibit osteogenesis and induce cell apoptosis via IFN-γ, thereby contributing to inflammation-induced bone loss [56]. IFN-γ, an immune-derived cytokine, contributes to the differentiation of osteoclasts, osteoblasts, and bone marrow adipocytes [57]. However, the specific mechanisms through which IFN-γ regulates bone metabolism in the context of chronic thyroid immune dysfunction remain unclear.

IL1β, a pro-inflammatory cytokine implicated in nearly all autoimmune diseases, plays a key role in bone loss associated with chronic inflammatory conditions, including HT [58]. Levescot et al. (2021) demonstrated that IL-1β can induce regulatory T cells (Tregs) to express RANKL, thereby promoting osteoclastogenesis and accelerating bone loss [59].

Finally, TNF-α affects both thyroid and bone health; its upregulation, triggered by either condition, can exacerbate the progression of the other. TNF-α directly damages thyroid follicular epithelial cells and inhibits osteoblast differentiation and bone formation [29,60].

Appendix A summarizes studies on the role of thyroid autoimmunity in bone health [29,44,45,46,48,51,54].

## 4. Impact of Oxidative Stress on Bone and Thyroid Health

Oxidative stress and inflammation are risk factors shared by Op and HT, and are closely interconnected processes. During the inflammatory response, the immune system produces high levels of reactive oxygen species (ROS) to eliminate pathogens. However, when ROS production becomes excessive and exceeds the body’s antioxidant defenses, it results in oxidative stress. Then, oxidative stress results from an imbalance between ROS and antioxidant agents [61], leading to cellular and molecular damage [62].

The relationship between oxidative stress and Op is well established [63,64,65]. A meta-analysis of 17 cross-sectional studies found that postmenopausal women with Op exhibited increased oxidative damage. Notably, superoxide dismutase (SOD) activity in plasma and serum samples showed a decreasing trend, highlighting a potential role of oxidative stress in the etiopathogenesis and clinical progression of Op [66]. Currently, the roles of different ROSs in bone metabolism are increasingly recognized. Zhao et al. (2021) found a cause–effect relationship between oxidant stress and osteoporosis in postmenopausal women, showing an imbalance between antioxidative and oxidative markers [67]. Hydrogen peroxide (H_2_O_2_), a byproduct of oxygen metabolism, suppresses mineralization by inhibiting osteoblast differentiation, inducing osteoblast apoptosis, and promoting osteoclast differentiation [68]. Superoxide anion, a first-generation ROS, negatively correlates with bone mass by increasing the RANKL/OPG ratio and enhancing bone resorption [69,70].

Several studies have also linked altered activity or reduced levels of antioxidant enzymes to an increased risk of Op. Glutathione peroxidase (GPx) mitigates oxidative damage by reducing organic hydroperoxides and H_2_O_2_, thereby preventing both osteoblast apoptosis and osteoclast activation [71]. Peroxiredoxin, when upregulated, inhibits osteoclast formation by decreasing RANKL levels, thus protecting against bone loss [72,73]. SOD catalyzes the conversion of superoxide radical to oxygen and H_2_O_2_. A deficiency in SOD is associated with bone loss, reduced osteoblastogenesis, and increased adipogenesis and osteoclastogenesis [74].

Thyroid function is also influenced by both inflammation and oxidative stress [75,76]. In HT, increased oxidative and inflammatory activity contributes to thyrocyte necrosis and apoptosis [76,77]. Excess H_2_O_2_ can damage DNA, induce lipid peroxidation, and lead to thyroid cell death [78]. Moreover, both inflammation and low TH levels impair the synthesis and activity of antioxidant enzymes [79,80]. Specifically, antioxidant enzymes (SOD and GPx) are reduced, resulting in an imbalance between total oxidant status and total antioxidant status [77]. As shown by Ates et al. (2015), oxidative stress varies across the different stages of HT, with the highest levels observed in individuals with overt disease (r −0.589) [77]. Riis et al. (2023) showed that levothyroxine treatment had no effects on the oxidative stress biomarkers, but the excretion of these latter was significantly higher in hypothyroid patients compared to healthy controls [81].

Appendix A summarizes studies on the role of oxidative stress in bone and thyroid health [67,75,76,77,81].

## 5. Impact of Dietary Habits on Bone and Thyroid Health

Given the central role of oxidative status in both bone and thyroid health, dietary habits may serve as a potential supportive strategy in the management of these disorders. Several studies have highlighted a possible link between dietary patterns, antioxidant intake, and overall health outcomes, suggesting that nutrition may influence the oxidative–inflammatory balance and thereby impact disease progression. High-fat diets and excessive sugar consumption overproduce ROS, increasing protein and lipid oxidation, DNA mutations, and inflammation [82,83,84,85]. Unhealthy dietary habits may promote inflammation and oxidative imbalance, two conditions characterizing both HT and Op. In their systematic review, Aleksandrova et al. (2021) demonstrated an inverse relationship between plant-based diets and oxidative stress [85].

Seafood, nuts, and wholegrain cereals are major dietary sources of selenium, an important agent that helps mitigate oxidative stress [86]. The recommended daily selenium intake ranges from 55 to 70 μg for adults, a level often not reached, especially in Europe and China [87]. Recent studies suggest that selenium deficiency could exacerbate the HT process, leading to progression to overt hypothyroidism [88]. In a prospective study with a follow-up of 6 years, subjects with low selenium levels and without selenium supplementation showed higher TPOAb levels and a higher rate of Hashimotos’s thyroiditis [relative risk 3.65; 95% C (1.03–12.90)] [89]. Although selenium supplementation is not recommended in the guidelines for the prevention and treatment of hypothyroidism or HT by the American and European Thyroid Associations [90,91], some evidence demonstrates a positive association between dietary selenium intake and femoral neck BMD in postmenopausal women, also after the adjustment for age, but not for other confounding variables (body mass index, waist circumference, race, education, income, alcohol consumption, coffee consumption, smoking status, clinic site, hormone use, thiazide use, thyroid medication use, physical activity, total caloric energy, dietary fat, dietary saturated fat, dietary protein, dietary magnesium, total calcium, total vitamin D, and other antioxidants) [92]. Xue et al. (2022) [93] showed that dietary selenium intake presented an inverted U-shape trend in relation to BMD levels, suggesting that the increasing dietary selenium intake is directly associated with BMD in individuals with low selenium levels. Specifically, the highest consumption of selenium from diet were associated with higher bone mineral density (total femur, neck, trochanter, intertrochanter, and lumbar spine) compared to the lowest consumption (β = 0.014, β = 0.010, β = 0.011, β = 0.017, and β = 0.013, respectively) [93]. Furthermore, Zhou et al. (2024) showed that a diet rich in antioxidant agents, especially selenium, was associated with a low risk of osteoporosis [OR 0.673, 95% CI (0.503–0.899)] [94]. Different enzymes, also involved in bone health, are affected by selenium levels. In particular, GPx has a central role in osteoclast by inhibiting osteoclastogenesis [95], and is poorly expressed when selenium is low, but restored with selenium supplementation [96]. A single nucleotide polymorphism at codon 198 of *GPx1* has been linked to low BMD and high bone resorption markers [97]. Selenium treatment also protects bone marrow stromal cells from the hydrogen-peroxide-induced suppression of osteoblastic differentiation by reducing oxidative stress [98]. Conversely, when selenoprotein expression falls, ROS levels rise, leading to excessive signaling and increased osteoclast activity, which damages bone microarchitecture [99]. Regarding thyroid function and the effectiveness of selenium supplementation, two randomized controlled trials have reported significant results, despite minor limitations. Yu et al. (2017) [100], in a study involving 60 HT patients, demonstrated that levothyroxine combined with selenium was more effective than levothyroxine alone in slowing disease progression. This may be attributed to the role of selenium-dependent enzymes, such as GPx and iodothyronine deiodinase, which are involved in antioxidant defense and thyroid hormone metabolism. Selenium supplementation was associated with reductions in thyroid autoantibody levels and inflammatory markers [100]. Similarly, a randomized controlled trial conducted by Krysiak et al. (2011) [101] confirmed the antioxidant and anti-inflammatory benefits of selenium when used alongside levothyroxine in HT patients. The proposed mechanism involves the normalization of monocyte function through levothyroxine and a reduction in pro-inflammatory cytokine release mediated by selenium [101].

Iodine is an essential nutrient for thyroid health and TH production [102]. Iodized salt, milk, fruits, and seafood are the primary dietary sources of iodine [103]. To prevent iodine deficiency, the World Health Organization (WHO) recommends fortifying salt with iodine at a concentration of 20 to 40 mcg/g. This range ensures adequate iodine intake when daily salt consumption is lower than 5 g/day [104]. Both chronic excess and deficiency of iodine can cause thyroid dysfunction, including thyroid autoimmunity. Indeed, highly iodinated thyroglobulin induces immune and autoimmune reactions [105]. Excess iodine reduces H_2_O_2_ due to the inhibition of the DUOX1/2 expression (Wollf–Chaikoff effect). Moreover, high doses of iodine could lead to lipid peroxidation [83]. Zhang et al. (2024) [106] demonstrated that chronic iodine excess could cause microarchitectural changes in bone. In particular, iodine induces modification in osteoblast and osteoclast activities, inhibiting both bone formation and resorption [106].

Beyond the intake of individual micro- and macronutrients, overall dietary patterns can exert a substantial impact on health. Among the most widely studied and adopted patterns, the Western diet (WD) and the Mediterranean diet (MD) represent two contrasting models with opposing effects on both thyroid and bone metabolism. While WD is typically associated with pro-inflammatory and oxidative stress-promoting properties, MD is rich in anti-inflammatory and antioxidant components, potentially offering protective effects for both endocrine and skeletal systems.

WD is characterized by a higher consumption of pre-packaged foods, refined grains, red and processed meats, sugary beverages, candies, high-fat dairy, and high-fructose products [107], with a generally lower intake of fruits, vegetables, legumes, whole grains, and nuts, which are essential sources of vitamins, minerals, fiber, and antioxidants [108]. It lacks antioxidants, such as vitamins C, E, and beta-carotene, which are essential for neutralizing ROS and preventing oxidative damage to lipids, proteins, and DNA [109,110,111,112,113]. On the other hand, WD is rich in pro-oxidant compounds that increase the production of ROS and oxidative stress [114]. The high consumption of saturated fats, processed foods, and refined sugars is associated with persistent low-grade inflammation [115]. WD is linked to several chronic diseases due to an increase in oxidative stress [116]. Different studies have shown that WD is associated with elevated levels of C-reactive protein (CRP), IL6, and TNF-α [117,118]. This chronic inflammation can lead to several chronic diseases, including HT [119]. Another important aspect is the impact of WD on the gut microbiome balance, leading to dysbiosis. This condition triggers systemic inflammation by altering immune cell differentiation and activation, cytokine production, immunoglobulin secretion, and immune tolerance [120]. The high content of saturated fats and cholesterol may trigger or exacerbate HT [121]. In their cross-sectional study, Henjum et al. (2023) [122] enrolled 205 subjects: 115 vegans, 55 lacto-ovo vegetarians, and 35 pescatarians. A positive correlation between animal fat and the levels of TPOAb or TgAb was found, and vegans had higher thyroglobulin levels compared to pescatarians [18 (10, 36) vs. 11 (5, 20) mcg/L] [122]. Furthermore, meat from animals raised with hormones may disrupt thyroid function [123,124].

MD represents the traditional dietary habits of the Mediterranean region, characterized by a high consumption of vegetables, fruits, legumes, whole grains, and nuts. Extra-virgin olive oil is the main fat used in MD, rich in essential dietary fatty acids, fat-soluble vitamins, phytosterols, and polyphenols [125,126]. This diet offers anti-inflammatory and antioxidant benefits [127,128]. Indeed, MD is recognized as a preventive tool against inflammatory, chronic, and degenerative diseases, reducing inflammation markers and improving endothelial disfunction. Chrysohoou et al. (2004) demonstrated that subjects with high adherence to MD have lower plasma concentrations of CRP, IL6, homocysteine, and fibrinogen, as well as a lower white blood cell count and a decrease in TNFα [129]. Lopez-Garcia et al. (2004) [130] showed a link between dietary patterns and markers of inflammation and endothelial dysfunction, including CRP, E-selectin, IL6, intercellular adhesion molecule (ICAM)-1, and vascular cell adhesion molecule (VCAM)-1. Indeed, subjects with higher adherence to MD have lower levels of CRP and E-selectin, while subjects with a higher intake of red meat, sweets, fries, and refined grains have higher levels of CRP, IL6, E-selectin, ICAM-1 and VCAM-1 concentrations [130]. Also, the randomized trial PREDIMED (Prevention with Mediterranean Diet) observed that a higher adherence to MD is linked to a significant reduction in cellular and serum inflammatory parameters [131]. Finally, HT subjects with high adherence to MD showed a better thyroid profile, suggesting that an anti-inflammatory diet is effective in managing HT [132,133]. Ruggeri et al. (2021) found that HT subjects had higher intake frequencies of animal-based foods and increased Advanced Glycation End products levels compared to controls (154.68 vs. 101.78 AU/g prot) [133]. Moreover, Ülker et al. (2023) [132] reported that TSH levels were lower in subjects following a gluten-free diet than controls (1.47 uIU/mL vs. 2.56 uIU/mL). Additionally, TPOAb and TgAb levels decreased in subjects following MD (70.76 vs. 68.95 IU/mL and 20.0 vs. 13.74 IU/mL, respectively) and a Mediterranean gluten-free diet (257.56 vs. 140.20 IU/mL and 32.66 vs. 6.31, respectively) compared to controls [132].

Appendix A summarizes studies on the impact of dietary habits and vitamin D on bone and thyroid health [43,85,89,92,93,94,100,101,108,113,117,122,129,130,131,132,133].

## 6. Hypovitaminosis D, Osteoporosis, and Hashimoto’s Thyroiditis

The vitamin D system is involved in calcium-phosphate balance and bone homeostasis, but also regulates several signaling pathways, cell proliferation, differentiation, apoptosis, and inflammation [134]. The vitamin D system plays a key role in immunomodulation, both in the innate and adaptive system [135] and is involved in controlling redox balance as antioxidant agent [136]. Several studies have reported the role of the vitamin D system in the onset of autoimmune diseases, such as HT [43,134,137]. Bozkurt et al. (2013) [138] showed a direct relationship between vitamin D and long-standing HT, as well as a positive relationship between vitamin D deficiency and the concentrations of antithyroid antibodies. Aktaş et al. (2020) found that low vitamin D levels in HT patients were associated with progression to hypothyroidism [139]. In addition, Filipova et al. (2023) [140] suggested that lower vitamin D levels exacerbated the severity of HT. Vitamin D supplementation appears to be effective in reducing circulating levels of TPOAb and TgAb in subjects with vitamin D deficiency [141,142,143]. Robat-Jazi et al. (2022), in a randomized controlled study, proved that subjects treated with cholecalciferol 50,000 UI showed lower INF-γ and TNF- α than the placebo group [144]. Despite this evidence, the relationship between vitamin D and HT remains inconsistent. Indeed, neither Sarmiento-Ramón et al. (2022) nor Demircioglu et al. (2021) found an association between hypovitaminosis D and HT [145,146]. In a meta-analysis conducted by Jiang et al. (2022), seven randomized controlled trials were conducted and no significant association was found between vitamin D treatment and TGAb, TSH, FT3, or FT4 levels. [147].

Appendix A summarizes studies on the impact of dietary habits and vitamin D on bone and thyroid health [138,139,141,142,143,144,145,146,147].

Adequate intake or population reference intake and foods with a higher amount of antioxidant agents are reported in Table 1 and Table 2, respectively.

## 7. Conclusions

Clinical and experimental evidence suggest that inflammation, oxidative stress, and dietary factors play a role in the development of thyroid and bone disorders. Additionally, TSH, FT3, and FT4 levels, as well as thyroid inflammation independently of thyroid function, appear to influence bone health. This aspect makes bone an important target in individuals affected by HT. Indeed, HT and Op share several risk factors, particularly oxidative stress and inflammation. Although it is difficult to present unequivocal conclusions, a correct dietary pattern rich in anti-inflammatory and antioxidant agents, such as vitamins A, C, D, and E, zinc, selenium, and iodine, can play a pivotal role in protection against mechanisms underlying the pathogenesis of both HT and Op. MD, with its high provision of fruits, vegetables, and whole grains and its low levels of meat and ultra-processed foods, has been shown to significantly reduce levels of chronic inflammation. The evidence from our review supports current dietary guidelines encouraging adherence to a high-quality diet that could be a winning strategy against both HT and Op.

Further studies are needed to better define the central role of thyroid disfunctions and to highlight the benefits of a high-quality diet and plant-based diet consumption in reducing inflammation and related chronic disease.

## Figures and Tables

**Table 1 nutrients-17-02109-t001:** Antioxidant agent dietary reference values (μg/day) for the European adult population.

	Male	Female	Pregnant Women	Lactating Women
Vitamin D (AI ^1^)	15	15	15	15
Selenium (AI ^1^)	70	70	70	85
Iodine (PRI ^2^)	150	150	200	200

^1^ AI: adequate intake; ^2^ PRI: population reference intake. [https://multimedia.efsa.europa.eu/drvs/index.htm, accessed on 21 June 2025].

**Table 2 nutrients-17-02109-t002:** Foods with higher amount of antioxidant agents.

Food	
	Vitamin D (μg/100 g)
Herring	45
Anchovy	16.5
Mackerel	4.4
Mushrooms	4.2
Mullet	2
	Selenium (μg/100 g)
Bovine kidney	145
Tuna	112
Bream	102
Sardines, fried	84.5
Lobster, boiled	68
Clam	64.6
Mussel	49
	Iodine (μg/100 g)
Seaweed	1000–8000
Cod	256
Salmon	100–200
Shrimp	100
Mussel	70

Data are expressed as μg of vitamin D (ergocalciferol and cholecalciferol) in 100 g of product, μg of selenium in 100 g of product, and μg of iodine in 100 g of product. [https://www.alimentinutrizione.it/sezioni/tabelle-nutrizionali, accessed on 21 June 2025].

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
