# Peer review of "Are Dietary Habits the Missing Link Between Hashimoto’s Thyroiditis and Osteoporosis?"

_nutrients, 2025, doi:10.3390/nu17132109_

Round 1
Reviewer 1 Report
Comments and Suggestions for Authors
This manuscript addresses an important and clinically relevant topic: the potential links between Hashimoto’s Thyroiditis (HT), osteoporosis, and dietary habits. The review is well written, logically structured, and largely comprehensive. However, there are several aspects that could be improved to enhance its scientific depth and clinical impact.
- Clarify the clinical significance of Hashimoto’s Thyroiditis on bone health
The manuscript would benefit from a more explicit description of the clinical burden of HT on skeletal health.
- Update and expand the bibliography
Although the references cited are appropriate, the manuscript would benefit from a more comprehensive and up-to-date literature base. Several recent and relevant studies—particularly those examining the association between thyroid autoimmunity and bone outcomes—are not included. For instance:
Soda M, Priante C, Pesce C, De Maio G, Lombardo M. The Impact of Vitamin D on
Immune Function and Its Role in Hashimoto's Thyroiditis: A Narrative Review.
Life (Basel). 2024 Jun 17;14(6):771.
Wu J, Li J, Yan Z, Yu X, Huang H. Higher prevalence of thyroid-specific autoantibodies (TPOAb and TgAb) is related to a higher prevalence of fractures in females: results from NHANES 2007–2010. Osteoporos Int. 2024;35(7):1213–1221.
Shi G, Lin Z, et al. Multiple thyroid disorders and risk of osteoporosis: a two-sample Mendelian randomization study. J Bone Miner Metab. 2025;43(2):96–107.
Incorporating such references would improve the scientific currency of the review and provide a more complete picture of current evidence.
- Improve manuscript structure
Section 1 contains introductory material and would be more appropriately titled “Introduction.” We suggest renaming it accordingly and adjusting the numbering of the subsequent sections to ensure clarity and coherence.
Author Response
Reviewer: 1
Comment 1: This manuscript addresses an important and clinically relevant topic: the potential links between Hashimoto’s Thyroiditis (HT), osteoporosis, and dietary habits. The review is well written, logically structured, and largely comprehensive.
Response 1: Thank you very much for your comment. All your suggestions have been carefully considered and incorporated into the manuscript.
Comment 2: However, there are several aspects that could be improved to enhance its scientific depth and clinical impact.
Clarify the clinical significance of Hashimoto’s Thyroiditis on bone health
The manuscript would benefit from a more explicit description of the clinical burden of HT on skeletal health.
Response 2: As you suggested, comments have been added to the text to clarify how thyroid health, and in particular Hashimoto’s thyroiditis, impact bone health.
“Based on this knowledge, Op and other osteo-metabolic disorders should be considered clinical conditions also affected by immune system disfunction [9]. Chronic inflammation accelerates bone loss and increases fracture risks, underscoring the detrimental effects of prolonged immune activation on skeletal integrity [10].
Hashimoto’s thyroiditis (HT) represents a prototypical autoimmune disorder, with an immunological phenotype that could affect skeletal development and bone homeostasis.” (page 2; lines 50-55).
“Given the critical role of the thyroid in bone homeostasis [15, 16] and the significant socio-economic impact of both conditions, the aim of this review is to explore and clarify the influence of HT and its related components on bone health, and to evaluate the potential role of dietary habits as a possible, yet often overlooked, link between these two pathological states” (page 2; lines 69-73)
Comment 3: Update and expand the bibliography
Although the references cited are appropriate, the manuscript would benefit from a more comprehensive and up-to-date literature base. Several recent and relevant studies—particularly those examining the association between thyroid autoimmunity and bone outcomes—are not included. For instance:
Soda M, Priante C, Pesce C, De Maio G, Lombardo M. The Impact of Vitamin D on
Immune Function and Its Role in Hashimoto's Thyroiditis: A Narrative Review.
Life (Basel). 2024 Jun 17;14(6):771.
Wu J, Li J, Yan Z, Yu X, Huang H. Higher prevalence of thyroid-specific autoantibodies (TPOAb and TgAb) is related to a higher prevalence of fractures in females: results from NHANES 2007–2010. Osteoporos Int. 2024;35(7):1213–1221.
Shi G, Lin Z, et al. Multiple thyroid disorders and risk of osteoporosis: a two-sample Mendelian randomization study. J Bone Miner Metab. 2025;43(2):96–107.
Incorporating such references would improve the scientific currency of the review and provide a more complete picture of current evidence.
Response 3: Thank you for your comment. As you suggested, the reference list has been expanded and updated, including the studies you recommended.
Comment 4: Improve manuscript structure
Section 1 contains introductory material and would be more appropriately titled “Introduction.” We suggest renaming it accordingly and adjusting the numbering of the subsequent sections to ensure clarity and coherence.
Response 4: The entire text has been revised following this comment. An introductory section has been added to clarify the context, as well as a dedicated section on oxidative stress to improve readability and understanding.
Reviewer 2 Report
Comments and Suggestions for Authors
I recommend major revisions:
-
Insufficient Critical Evaluation of Evidence. The manuscript largely compiles existing findings without consistently evaluating the quality or strength of the evidence. For instance, some key studies are mentioned without indicating study design, sample size, or limitations. I suggest including a comparative table summarizing study types (e.g., observational vs. interventional), key results, and conclusions.
-
Lack of Quantitative Data or Meta-analytical Insight. The review is descriptive and lacks numerical synthesis (e.g., odds ratios, relative risks, or prevalence rates). Including such data would improve the scientific robustness and practical relevance. Even if meta-analysis is not possible, a tabulated summary with numeric findings could enhance clarity.
-
Mechanistic Pathways Need Clarification. The manuscript refers to TH/TSH effects and cytokine involvement, but these mechanisms are not deeply explored. Figures or pathway diagrams would be valuable for explaining how HT influences bone metabolism, especially in euthyroid patients. The interplay between immune factors and bone homeostasis should be elaborated with more mechanistic detail.
-
Terminological Ambiguities and Overstatements. At times, the text uses terms like “well established” or “clear association” without robust backing. Please be cautious and ensure that all such claims are supported with specific references or tempered where data is preliminary.
-
HT in Euthyroid vs. Hypothyroid States. The central claim revolves around euthyroid HT, but many cited studies involve overt thyroid dysfunction. The distinction must be consistently maintained. If data are lacking for euthyroid patients, this gap should be transparently stated.
-
The title could be more informative. Consider rephrasing it as a question or clarifying the euthyroid focus (e.g., “Does Euthyroid Hashimoto’s Thyroiditis Affect Bone Health?”).
-
Improve flow between sections. Transitions between immunological, hormonal, and clinical subsections feel abrupt.
-
Some figure legends lack sufficient explanation; ensure all acronyms and pathways are well defined.
-
Revise the English for style and clarity in some sentences (e.g., lines 57–60).
-
Consider citing recent reviews/meta-analyses from 2022–2024 for up-to-date context.
Author Response
Reviewer: 2
I recommend major revisions:
Insufficient Critical Evaluation of Evidence. The manuscript largely compiles existing findings without consistently evaluating the quality or strength of the evidence. For instance, some key studies are mentioned without indicating study design, sample size, or limitations. I suggest including a comparative table summarizing study types (e.g., observational vs. interventional), key results, and conclusions.
We apologize for not having provided a thorough analysis of the studies included in the text. Therefore, several sentences and four supplementary tables have been added to assess the study design, the sample size of the examined population, the main findings, and the key limitations.
“Mazziotti et al. (2010) demonstrated that low-normal TSH levels were associated with a high prevalence of vertebral fractures (35%) in post-menopausal women, inde-pendently of THs, age and BMD. Also, high-normal TSH levels appear to be associated with a reduced risk of non-vertebral fractures (reduced by 35%) in post-menopausal women over a 6-year follow-up [18, 19]. Leader et al. (2014) confirmed that low TSH levels, even within the normal range (0.5–5.5 mIU/L), were associated with an in-creased risk of hip fractures in women [Odds Ratio (OR) 1.28, 95% CI (1.03-1.59)] but not in men [20]. In contrast, another study found no significant association between TSH levels within the normal range and changes in BMD, suggesting that TSH may not directly influence bone mass in euthyroid individuals, also after adjustment for height, weight, age, smoking, physical activity, and for postmenopausal females, use of hor-monal replacement therapy [21]. These conflicting findings point to potential size and sex-specific effects.
To summarize the role of TSH on bone health, Segna et al. demonstrated that individ-uals with subclinical hyperthyroidism experienced greater femoral neck bone loss (-0.18 %ΔBMD) compared to those with euthyroid function, whereas subclinical hy-pothyroidism was not associated with bone loss [22]. Bauer et al. (2001) reported an increased risk of hip [relative hazard (RH) 3.6] and vertebral fractures (OR 4.5, 95% CI 1.3, 15.6) in women over 65 years of age with low serum TSH levels, independently of hyperthyroidism [23]. Abrahamsen et al. (2014) found a higher risk of hip fractures and all main osteoporotic fractures in individuals with low TSH and normal T4 and T3 [24]. Finally, a meta-analysis including 70,298 subjects showed that subclinical hyper-thyroidism was associated with an increased risk of hip [Hazard Ratio (HR) 1.61] and other fractures (HR 1.98), particularly among those with TSH levels below 0.10 mIU/L pr with subclinical hypothyroidism [25]”. (pages 2, 3; lines 80-104).
“Van der Deure et al. (2008) demonstrated that free T4 (FT4) was negatively associated with bone parameters and BMD: lumbar spine BMD β -0.003, femoral neck BMD β -0.005, and cortical thickness β -0.001 [37]. Murphy et al. (2010) confirmed these re-sults, showing that higher FT4 (β -0.091) and FT3 (β -0.087) were associated with lower BMD at the hip, and higher FT4 was associated with increased bone loss at the hip (β -0.09), also after adjustment for age, body mass index, and BMD. Moreover, the risk of non-vertebral fracture was increased by 20% and 33% in women with higher FT4 or FT3, respectively. On the contrary, it was reduced by 35% in case of higher TSH [19]. To clarify the causal relationship between thyroid disorders and Op, Liu et al. (2025) used a two-sample Mendelian randomization analysis proving that TSH mediated 5.314% of the association between hypothyroidism and Op, while FT4 mediated 9.670% of the relationship between hyperthyroidism and Op [38]. In contrast, Leng et al. (2024) investigated the causal relationships between hypothyroidism, hyperthy-roidism, FT3, FT4, TSH, and OP risk, and found no significant causal associations be-tween FT3, FT4, TSH, and the risk of developing osteoporosis. Whereas, hypothyroid-ism may elevate the risk of osteoporosis (OR 1.082) by altering blood metabolite levels, such as triglycerides [39]”. (pages 3, 4; lines 123-139).
“Lambrinoudaki et al. (2017) evaluated the association between TPOAb and TgAb with BMD in euthyroid postmenopausal women demonstrating a higher risk of fractures in individuals with TPOAb and TgAb (OR 3.308 and 6.328, respectively) [44]. Polovina et al. (2017) showed that TPOAb is a potential marker of higher fracture risk in euthyroid postmenopausal women. TPOAb were found to be strongly linked with increased risk of any type of fractures, independently of TSH values (OR 7.8) [45]. Indeed, elevated TPOAb levels have been associated with higher Fracture Risk Assessment Tool (FRAX) score, indicating an increased risk of fractures. In particular, the FRAX score was sig-nificantly higher in the group with subclinical hypothyroidism than in the controls (6.5 vs 4.35) [46].
IL-6 seems to accelerate bone turnover by inhibiting bone formation and promoting bone reabsorption [47]. In particular, a meta-analysis conducted by Chen et al. showed a higher Op risk was in IL-6 572C/G additive (OR 2.25), dominant (OR 1.42) and reces-sive (OR 1.96) model [48]. Moreover, IL-6 produced increases RANKL mRNA expres-sion [49] and influences bone mass through glycoprotein 130-STAT 1/3 signaling pathway [50]”. (page 4; lines 147-161).
“In a prospective study with a follow-up of 6 years, subjects with low selenium levels and without selenium supplementation showed higher TPOAb levels and a higher rate of Hashimotos’s Thyroiditis [relative risk 3.65; 95% C (1.03-12.90)] [85].”. (page 6; lines 236- 239).
“Xue et al. (2022) showed that dietary selenium intake presented an inverted U-shape trend in relation to BMD levels, suggesting that the increasing dietary selenium intake is directly associated with BMD in individuals with low selenium levels. Specifically, the highest consumption of selenium from diet were associated with higher bone mineral density (total femur, neck, trochanter, intertrochanter and lumbar spine) compared to the lowest consumption (β=0.014, β=0.010, β=0.011, β=0.017 and β=0.013, respectively) [89]. Furthermore, Zhou et al. (2024) provided that a diet rich in antioxidant agents, especially selenium, was associated with a low risk of osteoporosis [OR 0.673, 95% CI (0.503-0.899)] [90]. Regarding thyroid function and the effectiveness of selenium supplementation, two randomized controlled trials have reported significant results, despite minor limitations. Yu et al. (2017), in a study involving 60 HT patients, demonstrated that levothyroxine combined with selenium was more effective than levothyroxine alone in slowing disease progression. This may be attributed to the role of selenium-dependent enzymes, such as GPx and iodothyronine deiodinase, which are involved in antioxidant defense and thyroid hormone metabolism. Selenium supplementation was associated with reductions in thyroid autoantibody levels and inflammatory markers [91]. Similarly, a randomized controlled trial conducted by Krysiak et al. (2011) confirmed the antioxidant and anti-inflammatory benefits of selenium when used alongside levothyroxine in HT patients. The proposed mechanism involves the normalization of monocyte function through levothyroxine and a reduction in pro-inflammatory cytokine release mediated by selenium [92]”. (page 6; lines 248- 268).
“Finally, HT subjects with high adherence to MD showed a better thyroid profile, sug-gesting that an anti‐inflammatory diet is effective in managing HT [123, 124]. Ruggeri et al. (2021) found that HT subjects had higher intake frequencies of animal-based foods and increased Advanced Glycation End products levels compared to controls (154.68 vs 101.78 AU/g prot) [124]. Moreover, Ülker et al. (2023) reported that TSH levels were lower in subjects following a gluten‐free diet than controls (1.47 uIU/mL vs 2.56 uIU/mL). Additionally, TPOAb and TgAb levels decreased in subjects following MD (70.76 vs 68.95 IU/mL and 20.0 vs 13.74 IU/mL, respectively) and Mediterranean gluten-free diet (257.56 vs 140.20 IU/mL and 32.66 vs 6.31, respectively) compared to controls [123]”. (page 8; lines 327- 336).
Lack of Quantitative Data or Meta-analytical Insight. The review is descriptive and lacks numerical synthesis (e.g., odds ratios, relative risks, or prevalence rates). Including such data would improve the scientific robustness and practical relevance. Even if meta-analysis is not possible, a tabulated summary with numeric findings could enhance clarity.
In accordance with your comment, four supplementary tables have been added to assess the study design, the sample size of the examined population, the main findings, and the key limitations.
Mechanistic Pathways Need Clarification. The manuscript refers to TH/TSH effects and cytokine involvement, but these mechanisms are not deeply explored. Figures or pathway diagrams would be valuable for explaining how HT influences bone metabolism, especially in euthyroid patients. The interplay between immune factors and bone homeostasis should be elaborated with more mechanistic detail.
In accordance with your suggestions, the paragraphs concerning the role of TSH, thyroid hormones, and inflammatory cytokines have been revised and reorganized.
“Thyroid-stimulating hormone (TSH), also known as thyrotropin, is a pituitary glycoprotein hormone that stimulates the release of THs, primarily thyroxine (T4), which is then converted into triiodothyronine (T3) [17]. THS role in bone health remains debated. Different studies have analysed the effect of subclinical hyperthyroidism and hypothyroidism on bone mineral density (BMD). Mazziotti et al. (2010) demonstrated that low-normal TSH levels were associated with a high prevalence of vertebral fractures (35%) in post-menopausal women, independently of THs, age and BMD. Also, high-normal TSH levels appear to be associated with a reduced risk of non-vertebral fractures (reduced by 35%) in post-menopausal women over a 6-year follow-up [18, 19]. Leader et al. (2014) confirmed that low TSH levels, even within the normal range (0.5–5.5 mIU/L), were associated with an increased risk of hip fractures in women [Odds Ratio (OR) 1.28, 95% CI (1.03-1.59)] but not in men [20]. In contrast, another study found no significant association between TSH levels within the normal range and changes in BMD, suggesting that TSH may not directly influence bone mass in euthyroid individuals, also after adjustment for height, weight, age, smoking, physical activity, and for postmenopausal females, use of hormonal replacement therapy [21]. These conflicting findings point to potential size and sex-specific effects.
To summarize the role of TSH on bone health, Segna et al. demonstrated that individuals with subclinical hyperthyroidism experienced greater femoral neck bone loss (-0.18 %ΔBMD) compared to those with euthyroid function, whereas subclinical hypothyroidism was not associated with bone loss [22]. Bauer et al. (2001) reported an increased risk of hip [relative hazard (RH) 3.6] and vertebral fractures (OR 4.5, 95% CI 1.3, 15.6) in women over 65 years of age with low serum TSH levels, independently of hyperthyroidism [23]. Abrahamsen et al. (2014) found a higher risk of hip fractures and all main osteoporotic fractures in individuals with low TSH and normal T4 and T3 [24]. Finally, a meta-analysis including 70,298 subjects showed that subclinical hyperthyroidism was associated with an increased risk of hip [Hazard Ratio (HR) 1.61] and other fractures (HR 1.98), particularly among those with TSH levels below 0.10 mIU/L pr with subclinical hypothyroidism [25].
Demonstrating the independent role of TSH alterations on bone health is a challenge. Abe et al. (2003) provided evidence for direct effects of TSH on bone formation, and bone resorption. These effects were mediated by the TSH receptor (TSHR). Indeed, haplo-insufficient TSHR mice showed bone loss and osteopenia [26]. Baliram et al. demonstrated that the absence of TSH signaling increased bone loss. Indeed, TSHR-Knockout mice with hyperthyroidism had a higher bone resorption rate compared wild-type mice [27]. TSH seems to inhibit osteoclast formation and increase osteoblast differentiation [28, 29], even if the exact mechanisms are not fully understood.
THs play a main role in the skeletal growth and maturation [30]. Children with undiagnosed severe hypothyroidism show delayed skeletal development, defective endochondral ossification, and short stature [31]. In adulthood, excess THs cause high bone turnover-mediated bone loss, while deficiency causes low bone turnover-mediated bone loss due to reduced osteoclastic bone resorption and decreased osteoblastic activity [31, 32]. T3 stimulates direct and indirect osteoblast proliferation and differentiation through various growth factors and cytokines, and by activating MAPK signaling and Wnt pathway [33]. Moreover, T3 appears to have a role in osteoblast response to parathormone (PTH) by modulating the expression of the PTH- and PTH related peptide- receptor [34]. On the contrary, it is not clear whether T3 acts directly on osteoclasts or indirectly by stimulating osteoblasts, osteocytes, or other bone cells [35, 36]. Van der Deure et al. (2008) demonstrated that free T4 (FT4) was negatively associated with bone parameters and BMD: lumbar spine BMD β -0.003, femoral neck BMD β -0.005, and cortical thickness β -0.001 [37]. Murphy et al. (2010) confirmed these results, showing that higher FT4 (β -0.091) and FT3 (β -0.087) were associated with lower BMD at the hip, and higher FT4 was associated with increased bone loss at the hip (β -0.09), also after adjustment for age, body mass index, and BMD. Moreover, the risk of non-vertebral fracture was increased by 20% and 33% in women with higher FT4 or FT3, respectively. On the contrary, it was reduced by 35% in case of higher TSH [19]. To clarify the causal relationship between thyroid disorders and Op, Liu et al. (2025) used a two-sample Mendelian randomization analysis proving that TSH mediated 5.314% of the association between hypothyroidism and Op, while FT4 mediated 9.670% of the relationship between hyperthyroidism and Op [38]. In contrast, Leng et al. (2024) investigated the causal relationships between hypothyroidism, hyperthyroidism, FT3, FT4, TSH, and OP risk, and found no significant causal associations between FT3, FT4, TSH, and the risk of developing osteoporosis. Whereas, hypothyroidism may elevate the risk of osteoporosis (OR 1.082) by altering blood metabolite levels, such as triglycerides [39]”. (pages 2- 4; lines 76- 139).
“HT directly impacts bone health, independently of hormonal dysfunction [40, 41]. TgAb, TPOAb and some cytokines dysregulated in HT individuals, such as IL1β, IL6, IL17, IL23, Tumor Necrosis Factor (TNF)- α and IFN–γ, are key pathogenic factors contributing to BMD alterations [42, 43].
Lambrinoudaki et al. (2017) evaluated the association between TPOAb and TgAb with BMD in euthyroid postmenopausal women demonstrating a higher risk of fractures in individuals with TPOAb and TgAb (OR 3.308 and 6.328, respectively) [44]. Polovina et al. (2017) showed that TPOAb is a potential marker of higher fracture risk in euthyroid postmenopausal women. TPOAb were found to be strongly linked with increased risk of any type of fractures, independently of TSH values (OR 7.8) [45]. Indeed, elevated TPOAb levels have been associated with higher Fracture Risk Assessment Tool (FRAX) score, indicating an increased risk of fractures. In particular, the FRAX score was significantly higher in the group with subclinical hypothyroidism than in the controls (6.5 vs 4.35) [46].
IL-6 seems to accelerate bone turnover by inhibiting bone formation and promoting bone reabsorption [47]. In particular, a meta-analysis conducted by Chen et al. showed a higher Op risk was in IL-6 572C/G additive (OR 2.25), dominant (OR 1.42) and recessive (OR 1.96) model [48]. Moreover, IL-6 produced increases RANKL mRNA expression [49] and influences bone mass through glycoprotein 130-STAT 1/3 signaling pathway [50].
Additionally, high IL17 levels are associated with low BMD in postmenopausal women [51], but its role in osteoblastogenesis remains to be further defined. Zhang et al. (2017) reported that IL17 inhibited osteoblast differentiation and bone regeneration in rats [52]. In contrary, Kim et al. (2020) showed that IL17 stimulated osteoblast differentiation and osteoblast-dependent osteoclastogenesis in vitro [53].
The IL-23/IL-17 axis plays a significant role in the pathogenesis of HT [54] as well as in systemin bone loss [55]. IL23 can indirectly inhibit osteogenesis and induce cell apoptosis via IFN-γ, thereby contributing to inflammation-induced bone loss [56]. IFN-γ, an immune-derived cytokine, contributes to the differentiation of osteoclasts, osteoblasts and bone marrow adipocytes [57]. However, the specific mechanisms through which IFN-γ regulates bone metabolism in the context of chronic thyroid immune dysfunction remain unclear.
IL1β, a pro-inflammatory cytokine implicated in nearly all autoimmune diseases, plays a key role in bone loss associated with chronic inflammatory conditions, including HT [58]. Levescot et al. (2021) demonstrated that IL-1β can induce regulatory T cells (Tregs) to express RANKL, thereby promoting osteoclastogenesis and accelerating bone loss [59].
Finally, TNF-α affects both thyroid and bone health; its upregulation, triggered by either condition, can exacerbate the progression of the other. TNF-α directly damages thyroid follicular epithelial cells and inhibits osteoblast differentiation and bone formation [29, 60]”. (pages 4, 5; lines 143- 182).
Terminological Ambiguities and Overstatements. At times, the text uses terms like “well established” or “clear association” without robust backing. Please be cautious and ensure that all such claims are supported with specific references or tempered where data is preliminary.
Thanks for the suggestion. All terminological ambiguities and overstatements have been reviewed and eliminated unless supported by scientific evidence.
HT in Euthyroid vs. Hypothyroid States. The central claim revolves around euthyroid HT, but many cited studies involve overt thyroid dysfunction. The distinction must be consistently maintained. If data are lacking for euthyroid patients, this gap should be transparently stated.
Thank you for your comment. It allows us to clarify and focus our article on its claim. Hashimoto’s thyroiditis, regardless of hormonal alterations, impacts bone health, and more importantly, maintaining a state of euthyroidism for as long as possible by reducing inflammation is the most sensible strategy to lower the risk of osteoporosis. The bridge between these two disorders and a key tool in managing both is represented by healthy dietary habits. The Mediterranean diet, rich in nutrients and antioxidants, has the potential to regulate both oxidative stress and inflammatory processes. it can help prevent diseases and reduce the risk of progression, acting as a therapeutic approach.
We have modified “The role of Thyroid-stimulating Hormone and Thyroid Hormones in bone health” to better highlight all the studies used. We began by analyzing the studies that focus on TSH function independently of thyroid hormone alterations, and then moved on to those addressing the effects of overt hyperthyroidism or hypothyroidism.
“Thyroid-stimulating hormone (TSH), also known as thyrotropin, is a pituitary glycoprotein hormone that stimulates the release of THs, primarily thyroxine (T4), which is then converted into triiodothyronine (T3) [17]. THS role in bone health remains debated. Different studies have analysed the effect of subclinical hyperthyroidism and hypothyroidism on bone mineral density (BMD). Mazziotti et al. (2010) demonstrated that low-normal TSH levels were associated with a high prevalence of vertebral fractures (35%) in post-menopausal women, independently of THs, age and BMD. Also, high-normal TSH levels appear to be associated with a reduced risk of non-vertebral fractures (reduced by 35%) in post-menopausal women over a 6-year follow-up [18, 19]. Leader et al. (2014) confirmed that low TSH levels, even within the normal range (0.5–5.5 mIU/L), were associated with an increased risk of hip fractures in women [Odds Ratio (OR) 1.28, 95% CI (1.03-1.59)] but not in men [20]. In contrast, another study found no significant association between TSH levels within the normal range and changes in BMD, suggesting that TSH may not directly influence bone mass in euthyroid individuals, also after adjustment for height, weight, age, smoking, physical activity, and for postmenopausal females, use of hormonal replacement therapy [21]. These conflicting findings point to potential size and sex-specific effects.
To summarize the role of TSH on bone health, Segna et al. demonstrated that individuals with subclinical hyperthyroidism experienced greater femoral neck bone loss (-0.18 %ΔBMD) compared to those with euthyroid function, whereas subclinical hypothyroidism was not associated with bone loss [22]. Bauer et al. (2001) reported an increased risk of hip [relative hazard (RH) 3.6] and vertebral fractures (OR 4.5, 95% CI 1.3, 15.6) in women over 65 years of age with low serum TSH levels, independently of hyperthyroidism [23]. Abrahamsen et al. (2014) found a higher risk of hip fractures and all main osteoporotic fractures in individuals with low TSH and normal T4 and T3 [24]. Finally, a meta-analysis including 70,298 subjects showed that subclinical hyperthyroidism was associated with an increased risk of hip [Hazard Ratio (HR) 1.61] and other fractures (HR 1.98), particularly among those with TSH levels below 0.10 mIU/L pr with subclinical hypothyroidism [25].
Demonstrating the independent role of TSH alterations on bone health is a challenge. Abe et al. (2003) provided evidence for direct effects of TSH on bone formation, and bone resorption. These effects were mediated by the TSH receptor (TSHR). Indeed, haplo-insufficient TSHR mice showed bone loss and osteopenia [26]. Baliram et al. demonstrated that the absence of TSH signaling increased bone loss. Indeed, TSHR-Knockout mice with hyperthyroidism had a higher bone resorption rate compared wild-type mice [27]. TSH seems to inhibit osteoclast formation and increase osteoblast differentiation [28, 29], even if the exact mechanisms are not fully understood.
THs play a main role in the skeletal growth and maturation [30]. Children with undiagnosed severe hypothyroidism show delayed skeletal development, defective endochondral ossification, and short stature [31]. In adulthood, excess THs cause high bone turnover-mediated bone loss, while deficiency causes low bone turnover-mediated bone loss due to reduced osteoclastic bone resorption and decreased osteoblastic activity [31, 32]. T3 stimulates direct and indirect osteoblast proliferation and differentiation through various growth factors and cytokines, and by activating MAPK signaling and Wnt pathway [33]. Moreover, T3 appears to have a role in osteoblast response to parathormone (PTH) by modulating the expression of the PTH- and PTH related peptide- receptor [34]. On the contrary, it is not clear whether T3 acts directly on osteoclasts or indirectly by stimulating osteoblasts, osteocytes, or other bone cells [35, 36]. Van der Deure et al. (2008) demonstrated that free T4 (FT4) was negatively associated with bone parameters and BMD: lumbar spine BMD β -0.003, femoral neck BMD β -0.005, and cortical thickness β -0.001 [37]. Murphy et al. (2010) confirmed these results, showing that higher FT4 (β -0.091) and FT3 (β -0.087) were associated with lower BMD at the hip, and higher FT4 was associated with increased bone loss at the hip (β -0.09), also after adjustment for age, body mass index, and BMD. Moreover, the risk of non-vertebral fracture was increased by 20% and 33% in women with higher FT4 or FT3, respectively. On the contrary, it was reduced by 35% in case of higher TSH [19]. To clarify the causal relationship between thyroid disorders and Op, Liu et al. (2025) used a two-sample Mendelian randomization analysis proving that TSH mediated 5.314% of the association between hypothyroidism and Op, while FT4 mediated 9.670% of the relationship between hyperthyroidism and Op [38]. In contrast, Leng et al. (2024) investigated the causal relationships between hypothyroidism, hyperthyroidism, FT3, FT4, TSH, and OP risk, and found no significant causal associations between FT3, FT4, TSH, and the risk of developing osteoporosis. Whereas, hypothyroidism may elevate the risk of osteoporosis (OR 1.082) by altering blood metabolite levels, such as triglycerides [39]”. (pages 2- 4; lines 76- 139)
The title could be more informative. Consider rephrasing it as a question or clarifying the euthyroid focus (e.g., “Does Euthyroid Hashimoto’s Thyroiditis Affect Bone Health?”).
As previously stated, the focus of our study is related to dietary habits, therefore, the title has not been modified according to your suggestion.
Improve flow between sections. Transitions between immunological, hormonal, and clinical subsections feel abrupt.
Thank you for your suggestion. The entire manuscript has been rewritten and organized according to the recommendations provided by all the reviewers.
Some figure legends lack sufficient explanation; ensure all acronyms and pathways are well defined.
The figure has been removed and is now presented only as a graphical summary.
Revise the English for style and clarity in some sentences (e.g., lines 57–60).
According your suggestion the sentence was revised as follows:
“It affects about 160 million people worldwide [11-13], predominantly adult women, with a female-to-male ratio of 10:1 [11]. HT is characterized by lymphocytic infiltration, varying degrees of thyroid disfunction, circulating antibodies against thyroid antigens, i.e. autoantibodies against thyroglobulin (TgAb) and thyroid peroxidase (TPOAb), and thyroid enlargement (i.e. goiter)”. (page 2; lines 57- 62).
Consider citing recent reviews/meta-analyses from 2022–2024 for up-to-date context.
Thank you for your comment. As you suggested, the reference list has been expanded and updated.
Reviewer 3 Report
Comments and Suggestions for Authors
This review discusses the connections between diet, Hashimoto's thyroiditis, and osteoporosis. However, I find this article is not in-depth. The author does not present many original viewpoints but instead simply lists studies by scholars. The viewpoints are essential for a review.
Why does this review lack an introduction? Please add an introduction to provide the research background.
Line 32: Please remove the period after the heading.
Lines 39-40: What are RANKL and RANK? These are not commonly known concepts. What roles do they play in osteoclasts? How do they promote osteoclast activation? These points should be elaborated on.
An introduction is needed because it is unclear why you are studying osteoporosis and Hashimoto’s Thyroiditis. Furthermore, the connection between HT and osteoporosis remains unclear. Is it simply because osteoporosis represents structural changes in bones, while HT represents immune dysfunction, and studies have found that microstructural changes in bones are related to immune alterations? What evidence links these two diseases?
Line 62: Remove “]”.
What are the harms of HT? Does it only lead to hypothyroidism? Are there further consequences?
The data on osteoporosis and HT pathology do not seem to be the most recent. Are there any updated data?
Lines 80-83: This sentence does not seem to convey any significant information. I think you could summarize your observations on TSH and draw a conclusion. For example, lower TSH levels are generally associated with fracture risk in postmenopausal women. Then cite the examples mentioned in your article. After that, you could note that there are also exceptions and provide a few counterexamples.
Line 99: Change the period to a comma: 70,298.
Line 95: The study by Bauer et al. also serves as evidence that low TSH levels increase fracture risk. Why is this study placed here?
Lines 78-101: You list many studies, but I think you should identify the commonalities and differences among them. These commonalities and differences would form your viewpoint—the summarized patterns you propose.
Line 110: What’s this?
Lines 158-159: ROS are also influenced by antioxidant enzymes synthesized in the body. Therefore, the statement "oxidative stress is caused by an imbalance between ROS and dietary antioxidants" is not entirely accurate.
Figure 1 is overly simplistic and lacks significant value. Additionally, the font formatting needs to be adjusted.
Section 4 lacks logical flow. You should first discuss in detail the relationships between oxidative stress, inflammation, and Op and HT. Then, categorize the discussion of diet’s impact on oxidative stress and how antioxidant interventions affect Op and HT.
Author Response
Reviewer: 3
This review discusses the connections between diet, Hashimoto's thyroiditis, and osteoporosis. However, I find this article is not in-depth. The author does not present many original viewpoints but instead simply lists studies by scholars. The viewpoints are essential for a review.
Thank you for your suggestion. In accordance with your comments, we have reorganized the manuscript by providing a more in-depth analysis of the cited studies and, most importantly, by adding four supplementary tables that evaluate the study design, sample size, main findings, and key limitations. This allowed us to refine our perspective and align more closely with the true objective of the research: dietary habits as prevention tool and treatment for disorders based on inflammation and oxidative stress.
Why does this review lack an introduction? Please add an introduction to provide the research background.
Following your suggestion, an introduction section was added:
“Osteoporosis (Op) is the most common osteo-metabolic disorder worldwide, affecting over 200 million people [1]. The incident of Op increases with age, and its prevalence is higher in women than men [1, 2]. Op is associated with an imbalance between bone resorption and formation, leading to a progressive deterioration of bone microarchitecture and bone mechanical properties, ultimately increasing the risk of fragility fractures [3]. Recent clinical and experimental evidence highlights the complex interaction between the bone and immune systems [4]. The immune and skeletal systems influence each other during cell activation, proliferation, and cellular senescence [4]. Macrophages enhance osteoblastogenesis via interleukin 18 (IL18) [5], while T cells influence osteoclastogenesis through IL1, IL6, IL4 and interferon-γ (IFN- γ) [6, 7]. On the other side, the receptor activator of nuclear factor-κB (RANK)–RANK Ligand (RANKL)– osteoprotegerin (OPG) axis, a signaling pathway that plays a crucial role in bone metabolism, is also involved in the regulation of immune cells. RANKL, a member of the tumor necrosis factor-α (TNF-α) superfamily, typically binds to RANK expressed on osteoclasts contributing to bone resorption. RANKL–RANK signaling also promotes the activation and survival of dendritic cells and T-cells, enhancing the immune response, and takes a part in B-cell recruitment and follicle organization [8].
Based on this knowledge, Op and other osteo-metabolic disorders should be considered clinical conditions also affected by immune system disfunction [9]. Chronic inflammation accelerates bone loss and increases fracture risks, underscoring the detrimental effects of prolonged immune activation on skeletal integrity [10].
Hashimoto’s thyroiditis (HT) represents a prototypical autoimmune disorder, with an immunological phenotype that could affect skeletal development and bone homeostasis. HT represents the most common cause of hypothyroidism in developed countries with adequate dietary iodine intake (i.e. median urinary iodine excretion ≥ 100 g/L). It affects about 160 million people worldwide [11-13], predominantly adult women, with a female-to-male ratio of 10:1 [11]. HT is characterized by lymphocytic infiltration, varying degrees of thyroid disfunction, circulating antibodies against thyroid antigens, i.e. autoantibodies against thyroglobulin (TgAb) and thyroid peroxidase (TPOAb), and thyroid enlargement (i.e. goiter).
HT is a recognized multisistemic disorder, that affects several organ systems, including including the neurological, cardiovascular, dermatological, gastrointestinal, and musculoskeletal systems. A deeper understanding of the systemic nature of Hashimoto's disease may unlock innovative therapeutic strategies that move beyond standard hormone replacement, aiming instead to modulate the fundamental autoimmune mechanisms driving the disorder [14].
Given the critical role of the thyroid in bone homeostasis [15, 16] and the significant socio-economic impact of both conditions, the aim of this review is to explore and clarify the influence of HT and its related components on bone health, and to evaluate the potential role of dietary habits as a possible, yet often overlooked, link between these two pathological states”. (pages 1, 2; lines 33- 73)
Line 32: Please remove the period after the heading.
As your suggestion, the sentence was removed.
Lines 39-40: What are RANKL and RANK? These are not commonly known concepts. What roles do they play in osteoclasts? How do they promote osteoclast activation? These points should be elaborated on.
Thank you for the suggestion. A sentence was added to better clarify the role on RANK-RANL-OPG axis:
“On the other side, the receptor activator of nuclear factor-κB (RANK)–RANK Ligand (RANKL)– osteoprotegerin (OPG) axis, a signaling pathway that plays a crucial role in bone metabolism, is also involved in the regulation of immune cells. RANKL, a member of the tumor necrosis factor-α (TNF-α) superfamily, typically binds to RANK expressed on osteoclasts contributing to bone resorption. RANKL–RANK signaling also promotes the activation and survival of dendritic cells and T-cells, enhancing the immune response, and takes a part in B-cell recruitment and follicle organization [8]”. (page 2; lines 42- 49)
An introduction is needed because it is unclear why you are studying osteoporosis and Hashimoto’s Thyroiditis. Furthermore, the connection between HT and osteoporosis remains unclear. Is it simply because osteoporosis represents structural changes in bones, while HT represents immune dysfunction, and studies have found that microstructural changes in bones are related to immune alterations? What evidence links these two diseases?
Thank you for your comment. We have worked to improve the writing of the text by emphasizing more clearly the connection between the two conditions. Different studies were added in the text that prove this link:
“Mazziotti et al. (2010) demonstrated that low-normal TSH levels were associated with a high prevalence of vertebral fractures (35%) in post-menopausal women, independently of THs, age and BMD. Also, high-normal TSH levels appear to be associated with a reduced risk of non-vertebral fractures (reduced by 35%) in post-menopausal women over a 6-year follow-up [18, 19]. Leader et al. (2014) confirmed that low TSH levels, even within the normal range (0.5–5.5 mIU/L), were associated with an increased risk of hip fractures in women [Odds Ratio (OR) 1.28, 95% CI (1.03-1.59)] but not in men [20]. In contrast, another study found no significant association between TSH levels within the normal range and changes in BMD, suggesting that TSH may not directly influence bone mass in euthyroid individuals, also after adjustment for height, weight, age, smoking, physical activity, and for postmenopausal females, use of hormonal replacement therapy [21]. These conflicting findings point to potential size and sex-specific effects.
To summarize the role of TSH on bone health, Segna et al. demonstrated that individuals with subclinical hyperthyroidism experienced greater femoral neck bone loss (-0.18 %ΔBMD) compared to those with euthyroid function, whereas subclinical hypothyroidism was not associated with bone loss [22]. Bauer et al. (2001) reported an increased risk of hip [relative hazard (RH) 3.6] and vertebral fractures (OR 4.5, 95% CI 1.3, 15.6) in women over 65 years of age with low serum TSH levels, independently of hyperthyroidism [23]. Abrahamsen et al. (2014) found a higher risk of hip fractures and all main osteoporotic fractures in individuals with low TSH and normal T4 and T3 [24]. Finally, a meta-analysis including 70,298 subjects showed that subclinical hyperthyroidism was associated with an increased risk of hip [Hazard Ratio (HR) 1.61] and other fractures (HR 1.98), particularly among those with TSH levels below 0.10 mIU/L pr with subclinical hypothyroidism [25]”. (pages 2, 3; lines 80- 104).
“Van der Deure et al. (2008) demonstrated that free T4 (FT4) was negatively associated with bone parameters and BMD: lumbar spine BMD β -0.003, femoral neck BMD β -0.005, and cortical thickness β -0.001 [37]. Murphy et al. (2010) confirmed these results, showing that higher FT4 (β -0.091) and FT3 (β -0.087) were associated with lower BMD at the hip, and higher FT4 was associated with increased bone loss at the hip (β -0.09), also after adjustment for age, body mass index, and BMD. Moreover, the risk of non-vertebral fracture was increased by 20% and 33% in women with higher FT4 or FT3, respectively. On the contrary, it was reduced by 35% in case of higher TSH [19]. To clarify the causal relationship between thyroid disorders and Op, Liu et al. (2025) used a two-sample Mendelian randomization analysis proving that TSH mediated 5.314% of the association between hypothyroidism and Op, while FT4 mediated 9.670% of the relationship between hyperthyroidism and Op [38]. In contrast, Leng et al. (2024) investigated the causal relationships between hypothyroidism, hyperthyroidism, FT3, FT4, TSH, and OP risk, and found no significant causal associations between FT3, FT4, TSH, and the risk of developing osteoporosis. Whereas, hypothyroidism may elevate the risk of osteoporosis (OR 1.082) by altering blood metabolite levels, such as triglycerides [39]”. (pages 3, 4; lines 123- 139).
“Lambrinoudaki et al. (2017) evaluated the association between TPOAb and TgAb with BMD in euthyroid postmenopausal women demonstrating a higher risk of fractures in individuals with TPOAb and TgAb (OR 3.308 and 6.328, respectively) [44]. Polovina et al. (2017) showed that TPOAb is a potential marker of higher fracture risk in euthyroid postmenopausal women. TPOAb were found to be strongly linked with increased risk of any type of fractures, independently of TSH values (OR 7.8) [45]. Indeed, elevated TPOAb levels have been associated with higher Fracture Risk Assessment Tool (FRAX) score, indicating an increased risk of fractures. In particular, the FRAX score was significantly higher in the group with subclinical hypothyroidism than in the controls (6.5 vs 4.35) [46].
IL-6 seems to accelerate bone turnover by inhibiting bone formation and promoting bone reabsorption [47]. In particular, a meta-analysis conducted by Chen et al. showed a higher Op risk was in IL-6 572C/G additive (OR 2.25), dominant (OR 1.42) and recessive (OR 1.96) model [48]. Moreover, IL-6 produced increases RANKL mRNA expression [49] and influences bone mass through glycoprotein 130-STAT 1/3 signaling pathway [50].
Additionally, high IL17 levels are associated with low BMD in postmenopausal women [51], but its role in osteoblastogenesis remains to be further defined. Zhang et al. (2017) reported that IL17 inhibited osteoblast differentiation and bone regeneration in rats [52]. In contrary, Kim et al. (2020) showed that IL17 stimulated osteoblast differentiation and osteoblast-dependent osteoclastogenesis in vitro [53].
The IL-23/IL-17 axis plays a significant role in the pathogenesis of HT [54] as well as in systemin bone loss [55]. IL23 can indirectly inhibit osteogenesis and induce cell apoptosis via IFN-γ, thereby contributing to inflammation-induced bone loss [56]. IFN-γ, an immune-derived cytokine, contributes to the differentiation of osteoclasts, osteoblasts and bone marrow adipocytes [57]. However, the specific mechanisms through which IFN-γ regulates bone metabolism in the context of chronic thyroid immune dysfunction remain unclear.
IL1β, a pro-inflammatory cytokine implicated in nearly all autoimmune diseases, plays a key role in bone loss associated with chronic inflammatory conditions, including HT [58]. Levescot et al. (2021) demonstrated that IL-1β can induce regulatory T cells (Tregs) to express RANKL, thereby promoting osteoclastogenesis and accelerating bone loss [59].
Finally, TNF-α affects both thyroid and bone health; its upregulation, triggered by either condition, can exacerbate the progression of the other. TNF-α directly damages thyroid follicular epithelial cells and inhibits osteoblast differentiation and bone formation [29, 60]”. (pages 4, 5; lines 147- 182)
Line 62: Remove “]”.
The typing was removed. Thank you.
What are the harms of HT? Does it only lead to hypothyroidism? Are there further consequences?????
Thank you for your comment, that allow us to address and explore this issue in different parts of the manuscript. In particular, as specified above, thyroiditis-induced changes in bone metabolism may also be independent of TSH value or even THs themselves.
“HT is a recognized multisistemic disorder, that affects several organ systems, including including the neurological, cardiovascular, dermatological, gastrointestinal, and musculoskeletal systems. A deeper understanding of the systemic nature of Hashimoto's disease may unlock innovative therapeutic strategies that move beyond standard hormone replacement, aiming instead to modulate the fundamental autoimmune mechanisms driving the disorder [14]”. (page 2; lines 63- 68)
“Thyroid-stimulating hormone (TSH), also known as thyrotropin, is a pituitary glycoprotein hormone that stimulates the release of THs, primarily thyroxine (T4), which is then converted into triiodothyronine (T3) [17]. THS role in bone health remains debated. Different studies have analysed the effect of subclinical hyperthyroidism and hypothyroidism on bone mineral density (BMD). Mazziotti et al. (2010) demonstrated that low-normal TSH levels were associated with a high prevalence of vertebral fractures (35%) in post-menopausal women, independently of THs, age and BMD. Also, high-normal TSH levels appear to be associated with a reduced risk of non-vertebral fractures (reduced by 35%) in post-menopausal women over a 6-year follow-up [18, 19]. Leader et al. (2014) confirmed that low TSH levels, even within the normal range (0.5–5.5 mIU/L), were associated with an increased risk of hip fractures in women [Odds Ratio (OR) 1.28, 95% CI (1.03-1.59)] but not in men [20]. In contrast, another study found no significant association between TSH levels within the normal range and changes in BMD, suggesting that TSH may not directly influence bone mass in euthyroid individuals, also after adjustment for height, weight, age, smoking, physical activity, and for postmenopausal females, use of hormonal replacement therapy [21]. These conflicting findings point to potential size and sex-specific effects.
To summarize the role of TSH on bone health, Segna et al. demonstrated that individuals with subclinical hyperthyroidism experienced greater femoral neck bone loss (-0.18 %ΔBMD) compared to those with euthyroid function, whereas subclinical hypothyroidism was not associated with bone loss [22]. Bauer et al. (2001) reported an increased risk of hip [relative hazard (RH) 3.6] and vertebral fractures (OR 4.5, 95% CI 1.3, 15.6) in women over 65 years of age with low serum TSH levels, independently of hyperthyroidism [23]. Abrahamsen et al. (2014) found a higher risk of hip fractures and all main osteoporotic fractures in individuals with low TSH and normal T4 and T3 [24]. Finally, a meta-analysis including 70,298 subjects showed that subclinical hyperthyroidism was associated with an increased risk of hip [Hazard Ratio (HR) 1.61] and other fractures (HR 1.98), particularly among those with TSH levels below 0.10 mIU/L pr with subclinical hypothyroidism [25].
Demonstrating the independent role of TSH alterations on bone health is a challenge. Abe et al. (2003) provided evidence for direct effects of TSH on bone formation, and bone resorption. These effects were mediated by the TSH receptor (TSHR). Indeed, haplo-insufficient TSHR mice showed bone loss and osteopenia [26]. Baliram et al. demonstrated that the absence of TSH signaling increased bone loss. Indeed, TSHR-Knockout mice with hyperthyroidism had a higher bone resorption rate compared wild-type mice [27]. TSH seems to inhibit osteoclast formation and increase osteoblast differentiation [28, 29], even if the exact mechanisms are not fully understood.
THs play a main role in the skeletal growth and maturation [30]. Children with undiagnosed severe hypothyroidism show delayed skeletal development, defective endochondral ossification, and short stature [31]. In adulthood, excess THs cause high bone turnover-mediated bone loss, while deficiency causes low bone turnover-mediated bone loss due to reduced osteoclastic bone resorption and decreased osteoblastic activity [31, 32]. T3 stimulates direct and indirect osteoblast proliferation and differentiation through various growth factors and cytokines, and by activating MAPK signaling and Wnt pathway [33]. Moreover, T3 appears to have a role in osteoblast response to parathormone (PTH) by modulating the expression of the PTH- and PTH related peptide- receptor [34]. On the contrary, it is not clear whether T3 acts directly on osteoclasts or indirectly by stimulating osteoblasts, osteocytes, or other bone cells [35, 36]. Van der Deure et al. (2008) demonstrated that free T4 (FT4) was negatively associated with bone parameters and BMD: lumbar spine BMD β -0.003, femoral neck BMD β -0.005, and cortical thickness β -0.001 [37]. Murphy et al. (2010) confirmed these results, showing that higher FT4 (β -0.091) and FT3 (β -0.087) were associated with lower BMD at the hip, and higher FT4 was associated with increased bone loss at the hip (β -0.09), also after adjustment for age, body mass index, and BMD. Moreover, the risk of non-vertebral fracture was increased by 20% and 33% in women with higher FT4 or FT3, respectively. On the contrary, it was reduced by 35% in case of higher TSH [19]. To clarify the causal relationship between thyroid disorders and Op, Liu et al. (2025) used a two-sample Mendelian randomization analysis proving that TSH mediated 5.314% of the association between hypothyroidism and Op, while FT4 mediated 9.670% of the relationship between hyperthyroidism and Op [38]. In contrast, Leng et al. (2024) investigated the causal relationships between hypothyroidism, hyperthyroidism, FT3, FT4, TSH, and OP risk, and found no significant causal associations between FT3, FT4, TSH, and the risk of developing osteoporosis. Whereas, hypothyroidism may elevate the risk of osteoporosis (OR 1.082) by altering blood metabolite levels, such as triglycerides [39]” (pages 2- 4; lines 76- 139)
The data on osteoporosis and HT pathology do not seem to be the most recent. Are there any updated data?
Thank you for your comment. As you suggested, the reference list has been expanded and updated.
Lines 80-83: This sentence does not seem to convey any significant information. I think you could summarize your observations on TSH and draw a conclusion. For example, lower TSH levels are generally associated with fracture risk in postmenopausal women. Then cite the examples mentioned in your article. After that, you could note that there are also exceptions and provide a few counterexamples.
According to your suggestion, the sentence was modified as follows:
“THS role in bone health remains debated. Different studies have analysed the effect of subclinical hyperthyroidism and hypothyroidism on bone mineral density (BMD)”. (page 2; lines 78- 80).
Line 99: Change the period to a comma: 70,298.
The typing error has been corrected as you suggested.
Line 95: The study by Bauer et al. also serves as evidence that low TSH levels increase fracture risk. Why is this study placed here?
Thank you for giving us the opportunity to clarify the choice we made. The study by Bauer and colleagues is a prospective study involving postmenopausal women and followed for 6 years. Women with TSH levels of 0.1 mIU/L or less had a significantly increased risk of new hip [relative hazard 3.6; 95% CI (1.0, 12.9)] and vertebral [OR 4.5, 95% CI (1.3, 15.6)] fractures], independently of age, hyperthyroidism, self-rated health, and use of estrogen and thyroid hormone, compare with subjects with TSH levels in the normal range (0.5-5.5 mU/L). Moreover, the risk of vertebral fracture was significantly elevated among women with borderline low TSH levels (0.1 to 0.5 mIU/L). In light of these results, we chose to place the study in this section of the manuscript.
“Bauer et al. (2001) reported an increased risk of hip [relative hazard (RH) 3.6] and vertebral fractures (OR 4.5, 95% CI 1.3, 15.6) in women over 65 years of age with low serum TSH levels, independently of hyperthyroidism [23].”. (page 3; lines 96- 99)
Lines 78-101: You list many studies, but I think you should identify the commonalities and differences among them. These commonalities and differences would form your viewpoint—the summarized patterns you propose.
In accordance with your comments, we have reorganized the manuscript by providing a more in-depth analysis of the cited studies and by adding four supplementary tables that evaluate commonalities and differences between them. This allowed us to refine our perspective and align more closely with the true objective of the research: dietary habits as prevention tool and treatment for disorders based on inflammation and oxidative stress.
Line 110: What’s this?
We apologize for the error.
Lines 158-159: ROS are also influenced by antioxidant enzymes synthesized in the body. Therefore, the statement "oxidative stress is caused by an imbalance between ROS and dietary antioxidants" is not entirely accurate.
I apologize for the superficial nature of the sentence, which has been corrected as follows:
“Then, oxidative stress results from an imbalance between ROS and antioxidant agents [61], leading to cellular and molecular damage [62]”. (page 5; lines 190, 191)
Figure 1 is overly simplistic and lacks significant value. Additionally, the font formatting needs to be adjusted.
The figure has been removed and is now presented only as a graphical summary.
Section 4 lacks logical flow. You should first discuss in detail the relationships between oxidative stress, inflammation, and Op and HT. Then, categorize the discussion of diet’s impact on oxidative stress and how antioxidant interventions affect Op and HT.
As you suggested, Section 4 has been modified to follow the logical flow of the discussion. Specifically, we have divided Section 4 into two distinct parts:
“4. Impact of oxidative stress on bone and thyroid health
Oxidative stress and inflammation are risk factors shared by both Op, and are closely interconnected processes. During the inflammatory response, the immune system produces high levels of reactive oxygen species (ROS) to eliminate pathogens. Howev-er, when ROS production becomes excessive and exceeds the body’s antioxidant de-fenses, it results in oxidative stress. Then, oxidative stress results from an imbalance between ROS and antioxidant agents [61], leading to cellular and molecular damage [62].
The relationship between oxidative stress and Op is well established [63]. A me-ta-analysis of 17 cross-sectional studies found that postmenopausal women with Op exhibited increased oxidative damage. Notably, superoxide dismutase (SOD) activity in plasma and serum samples showed a decreasing trend, highlighting a potential role of oxidative stress in the etiopathogenesis and clinical progression of Op [64]. Currently, the roles of different ROSs in bone metabolism are increasingly recognized. Hydrogen peroxide (H₂O₂), a byproduct of oxygen metabolism, suppresses mineraliza-tion by inhibiting osteoblast differentiation, inducing osteoblast apoptosis, and pro-moting osteoclast differentiation [65]. Superoxide anion, a first-generation ROS, nega-tively correlates with bone mass by increasing the RANKL/OPG ratio and enhancing bone resorption [66, 67].
Several studies have also linked altered activity or reduced levels of antioxidant en-zymes to an increased risk of Op. Glutathione peroxidase (GPx) mitigates oxidative damage by reducing of organic hydroperoxides and H2O2, thereby preventing both os-teoblast apoptosis and osteoclast activation [68]. Peroxiredoxin, when upregulated, in-hibits osteoclast formation by decreasing RANKL levels, thus protecting against bone loss [69, 70]. SOD catalyzes the conversion of superoxide radical to oxygen and H2O2. A deficiency in SOD is associated with bone loss, reduced osteoblastogenesis, and in-creased adipogenesis and osteoclastogenesis [71].
Thyroid function is also influenced by both inflammation and oxidative stress [72, 73]. In HT, increased oxidative and inflammatory activity contributes to thyrocyte necrosis and apoptosis [73, 74]. Excess H₂O₂ can damage DNA, induce lipid peroxidation, and lead to thyroid cell death [75]. Moreover, both inflammation and low THs levels im-pair the synthesis and activity of antioxidant enzymes [76, 77]. Specifically, antioxi-dant enzymes (SOD and GPx) are reduced resulting in an imbalance between total ox-idant status and total antioxidant status [74]. As showed by Ates et al. (2015), oxida-tive stress varies across the different stages of HT, with the highest levels observed in individuals with overt disease (r -0.589) [74].
Supplementary table 3 summarizes studies regarding the role oxidative stress on bone and thyroid health [72- 74].
- Impact of dietary habits on bone and thyroid health
Given the central role of oxidative status in both bone and thyroid health, dietary hab-its may serve as a potential supportive strategy in the management of these disorders. Several studies have highlighted a possible link between dietary patterns, antioxidant intake, and overall health outcomes, suggesting that nutrition may influence the oxi-dative-inflammatory balance and thereby impact disease progression. High-fat diets and excessive sugar consumption overproduce ROS, increasing protein and lipid oxi-dation, DNA mutations, and inflammation [78- 80]. Unhealthy dietary habits may promote inflammation and oxidative imbalance, two conditions featured by both HT and Op. In their systematic review, Aleksandroba et al. (2021) demonstrared an en-verse relationshiop between plant-based diets and oxidative stress [81].
Seafood, nuts and also wholegrain cereals are major dietary sources of selenium, an important agent that helps mitigate oxidative stress [82]. The recommended daily se-lenium intake ranges from 55 to 70 μg for adults, a level often not reached, especially in Europe and China [83]. Recent studies suggest that selenium deficiency could exac-erbate HT process, leading to progression to overt hypothyroidism [84]. In a prospec-tive study with a follow-up of 6 years, subjects with low selenium levels and without selenium supplementation showed higher TPOAb levels and a higher rate of Hash-imotos’s Thyroiditis [relative risk 3.65; 95% C (1.03-12.90)] [85]. Although selenium supplementation is not recommended in the guidelines for the prevention and treat-ment of hypothyroidism or HT by the American and European Thyroid Associations [86, 87], some evidence demonstrates a positive association between dietary selenium intake and femoral neck BMD in post-menopausal women, also after the adjustment for age, but not for other confounding variables (body mass index, waist circumfer-ence, race, education, income, alcohol consumption, coffee consumption, smoking sta-tus, clinic site, hormone use, thiazide use, thyroid medication use, physical activity, total caloric energy, dietary fat, dietary saturated fat, dietary protein, dietary magne-sium, total calcium, total vitamin D, and other antioxidants) [88]. Xue et al. (2022) showed that dietary selenium intake presented an inverted U-shape trend in relation to BMD levels, suggesting that the increasing dietary selenium intake is directly asso-ciated with BMD in individuals with low selenium levels. Specifically, the highest consumption of selenium from diet were associated with higher bone mineral density (total femur, neck, trochanter, intertrochanter and lumbar spine) compared to the lowest consumption (β=0.014, β=0.010, β=0.011, β=0.017 and β=0.013, respectively) [89]. Furthermore, Zhou et al. (2024) provided that a diet rich in antioxidant agents, especially selenium, was associated with a low risk of osteoporosis [OR 0.673, 95% CI (0.503-0.899)] [90]. Regarding thyroid function and the effectiveness of selenium sup-plementation, two randomized controlled trials have reported significant results, de-spite minor limitations. Yu et al. (2017), in a study involving 60 HT patients, demon-strated that levothyroxine combined with selenium was more effective than levothy-roxine alone in slowing disease progression. This may be attributed to the role of sele-nium-dependent enzymes, such as GPx and iodothyronine deiodinase, which are in-volved in antioxidant defense and thyroid hormone metabolism. Selenium supple-mentation was associated with reductions in thyroid autoantibody levels and inflam-matory markers [91]. Similarly, a randomized controlled trial conducted by Krysiak et al. (2011) confirmed the antioxidant and anti-inflammatory benefits of selenium when used alongside levothyroxine in HT patients. The proposed mechanism involves the normalization of monocyte function through levothyroxine and a reduction in pro-inflammatory cytokine release mediated by selenium [92].
Iodine is an essential nutrient for thyroid health and TH production [93]. Iodized salt, milk, fruits, and seafood are the primary dietary sources of iodine [94]. To prevent io-dine deficiency, the World Health Organization (WHO) recommends fortifying salt with iodine at a concentration of 20 to 40 mcg/g. This range ensures adequate iodine intake when daily salt consumption is lower than 5 g/day [95]. Both chronic excess and deficiency of iodine can cause thyroid dysfunction, including thyroid autoimmunity. Indeed, highly iodinated thyroglobulin induces immune and autoimmune reactions [96]. Excess iodine reduces hydrogen peroxide (H2O2) due to the inhibition of the DUOX1/2 expression (Wollf–Chaikoff effect). Moreover, high doses of iodine could lead to lipid peroxidation [79]. Zhang et al. (2024) demonstrated that chronic iodine excess could cause microarchitectural changes in bone. In particular, iodine induces modification in osteoblast and osteoclast activities, inhibiting both bone formation and resorption [97].
Beyond the intake of individual micro- and macronutrients, overall dietary patterns can exert a substantial impact on health. Among the most widely studied and adopted patterns, the Western Diet (WD) and the Mediterranean Diet (MD) represent two con-trasting models with opposing effects on both thyroid and bone metabolism. While the WD is typically associated with pro-inflammatory and oxidative stress-promoting properties, the MD is rich in anti-inflammatory and antioxidant components, poten-tially offering protective effects for both endocrine and skeletal systems.
WD is characterized by a higher consumption of pre-packaged foods, refined grains, red and processed meats, sugary beverages, candies, high-fat dairy and high-fructose products [98], with a generally lower intake of fruits, vegetables, legumes, whole grains, and nuts, which are essential sources of vitamins, minerals, fiber, and antioxi-dants [99]. It lacks antioxidants, such as vitamins C, E and beta-carotene, which are essential for neutralizing ROS and preventing oxidative damage to lipids, proteins, and DNA [100- 104]. On the other hand, WD is rich in pro-oxidant compounds that increase the production of ROS and oxidative stress [105]. The high consumption of saturated fats, processed foods, and refined sugars is associated with persistent low-grade inflammation [106]. WD is linked to several chronic diseases due to an in-creasing of oxidative stress [107]. Different studies have shown that WD is associated with elevated levels of C-reactive protein (CRP), IL6 and TNF-α [108, 109]. This chronic inflammation can lead to several chronic diseases, including HT [110]. Anoth-er important aspect is the impact of WD on the gut microbiome balance, leading to dysbiosis. This condition triggers systemic inflammation by altering immune cell dif-ferentiation and activation, cytokine production, immunoglobulin secretion, and im-mune tolerance [111]. The high content of saturated fats and cholesterol may trigger or exacerbate HT [112]. In his cross-sectional study Henjum et al. (2023) enrolled 205 subjects: 115 vegans, 55 lacto-ovo vegetarians and 35 pescatarians. A positive correla-tion between animal fat and the levels of TPOAb or TgAb was found and vegans had higher thyroglobulin levels compared to pescatarians [18 (10, 36) vs 11(5, 20) mcg/L] [113]. Furthermore, meat from animals raised with hormones may disrupt thyroid function [114, 115].
MD represents the traditional dietary habits of the Mediterranean region, character-ized by high consumption of vegetables, fruits, legumes, whole grains and nuts. Ex-tra-virgin olive oil is the main fat used in MD, rich in essential dietary fatty acids, fat-soluble vitamins, phytosterols, and polyphenols [116, 117]. This diet offers an-ti-inflammatory and antioxidant benefits [118, 119]. Indeed, MD is recognized as a preventive tool against inflammatory, chronic, and degenerative diseases, reducing in-flammation markers and improving endothelial disfunction. Chrysohoou et al. (2004) demonstrated that subjects with high adherence to MD have lower plasma concentra-tions of CRP, IL6, homocysteine and fibrinogen, as well as a lower white blood cell count, and a decrease of TNFα [120]. Lopez-Garcia et al. (2004) showed a link between dietary patterns and markers of inflammation and endothelial dysfunction, including CRP, E-selectin, IL6, Intercellular Adhesion Molecule (ICAM)-1, and Vascular Cell Aadhesion Molecule (VCAM)-1. Indeed, subjects with higher adherence to MD have lower levels of CRP and E-selectin, while subjects with a higher intake of red meat, sweets, fries, and refined grains have higher levels of with CRP, IL6, E-selectin, ICAM-1 and VCAM-1 concentrations [121]. Also, the randomized trial PREDIMED (Prevention with Mediterranean Diet) observed that a higher adherence to MD is linked to a significant reduction in cellular and serum inflammatory parameters [122]. Finally, HT subjects with high adherence to MD showed a better thyroid profile, sug-gesting that an anti‐inflammatory diet is effective in managing HT [123, 124]. Ruggeri et al. (2021) found that HT subjects had higher intake frequencies of animal-based foods and increased Advanced Glycation End products levels compared to controls (154.68 vs 101.78 AU/g prot) [124]. Moreover, Ülker et al. (2023) reported that TSH levels were lower in subjects following a gluten‐free diet than controls (1.47 uIU/mL vs 2.56 uIU/mL). Additionally, TPOAb and TgAb levels decreased in subjects following MD (70.76 vs 68.95 IU/mL and 20.0 vs 13.74 IU/mL, respectively) and Mediterranean gluten-free diet (257.56 vs 140.20 IU/mL and 32.66 vs 6.31, respectively) compared to controls [123].
Supplementary table 4 summarizes studies regarding the impact of dietary habits and vitamin D on bone and thyroid health [43, 81, 85, 88- 92, 99, 104, 108, 113, 120-124]”. (pages 5- 8; lines 185- 338)
Round 2
Reviewer 2 Report
Comments and Suggestions for Authors
Congratulations. I recommend to publish this manuscript
Reviewer 3 Report
Comments and Suggestions for Authors
After revisions, the quality of the manuscript has improved.
Introduction: Please add content related to oxidative stress and diet in the background.
Table 3: These three studies are outdated, though you may keep them. However, you also need to include some newer studies. After searching, I have come across some recent studies, such as those from 2022. Please add some of these newer studies to Table 3.
Section 5: You suggest that selenium supplementation may improve BMD by alleviating oxidative stress, but none of the studies you cited measured any oxidative stress-related indicators. Could you provide additional studies to support your argument?
Author Response
Comments 1: After revisions, the quality of the manuscript has improved.
Response 1: Thank you very much, the suggestions provided helped to clearly focus the central point of view and improve the entire work.
Comment 2: Introduction: Please add content related to oxidative stress and diet in the background.
Response 2: Thank you for the suggestion. In the introduction, a reference to both oxidative stress and dietary habits has been included in order to better introduce the topics discussed in the remaining sections of the review.
“Based on the available literature, it is evident that both Op and HT share common pathogenic mechanisms, such as oxidative stress and unhealthy dietary habits. On the one hand, oxidative stress, resulting from an imbalance between pro-oxidant and anti-oxidant agents, affects both bone and thyroid health. On the other hand, the influence of a healthy lifestyle, and in particular, an appropriate dietary patterns, may offer new insights into the pathophysiology of both disorders and contribute to the development of novel therapeutic strategies”. (page2; lines 69- 75)
Comment 3: Table 3: These three studies are outdated, though you may keep them. However, you also need to include some newer studies. After searching, I have come across some recent studies, such as those from 2022. Please add some of these newer studies to Table 3.
Response 3: Thank you for the suggestion. Four updated studies have been added to the manuscript, and in particular, two of these have been included in the supplementary table. For clarity, all supplementary tables contain the studies used in the review but refer only to those involving humans.
Marcucci, G.; Domazetovic, V.; Nediani, C.; Ruzzolini, J.; Favre, C.; Brandi, M.L. Oxidative Stress and Natural Antioxidants in Osteoporosis: Novel Preventive and Therapeutic Approaches. Antioxidants 2023, 12:373.
Zhao, H.; Yu, F.; Wu, W. New Perspectives on Postmenopausal Osteoporosis: Mechanisms and Potential Therapeutic Strategies of Sirtuins and Oxidative Stress. Antioxidants 2025, 14:605.
“Zhao et al (2021) found a cause-effect relationship between oxidant stress and osteoporosis in postmenopausal women, showing an imbalance between antioxidative and oxidative markers [67]”. (page 5; lines 204-207).
“Riis et al (2023) showed that levothyroxine treatment had no effects on the oxidative stress biomarkers, but the excretion of these latter was significantly higher in hypo-thyroid patients compared to healthy controls [81]”. (page 6; lines 228-230)
Comment 4: Section 5: You suggest that selenium supplementation may improve BMD by alleviating oxidative stress, but none of the studies you cited measured any oxidative stress-related indicators. Could you provide additional studies to support your argument?
Response 4: According to your suggestion, we have added a new sentence regarding the role of selenium on bone and oxidative stress.
“Different enzymes, involved also in bone health, are affected by selenium levels. In particular, GPx has a central role in osteoclast by inhibiting osteoclastogenesis [95], and is poorly expressed when selenium is low, but restored with selenium supplementation [96]. A single nucleotide polymorphism at codon 198 of GPx1 has been linked to low BMD and high bone resorption markers [97]. Selenium treatment also protects bone marrow stromal cells from the hydrogen-peroxide-induced suppression of osteoblastic differentiation by reducing oxidative stress [98]. Conversely, when selenoprotein expression falls, ROS levels rise, leading to excessive signaling and increased osteoclast activity, which damages bone microarchitecture [99]”. (pages 7, 6; lines 268-276).